# Structure and Properties of Zr-Mo-Si-B-(N) Hard Coatings Obtained by d.c. Magnetron Sputtering of ZrB_2_-MoSi_2_ Target

**DOI:** 10.3390/ma14081932

**Published:** 2021-04-13

**Authors:** Philipp Kiryukhantsev-Korneev, Alina Sytchenko, Yuriy Pogozhev, Stepan Vorotilo, Anton Orekhov, Pavel Loginov, Evgeny Levashov

**Affiliations:** 1Scientific-Educational Center of SHS, National University of Science and Technology “MISiS”, 119049 Moscow, Russia; alina-sytchenko@yandex.ru (A.S.); yspogozhev@mail.ru (Y.P.); stepan.vorotylo@gmail.com (S.V.); orekhov.anton@gmail.com (A.O.); pavel.loginov.misis@list.ru (P.L.); levashov@shs.misis.ru (E.L.); 2Laboratory of Electron Diffraction, Federal Scientific Research Centre “Crystallography and Photonics”, Russian Academy of Sciences, Leninsky Prospect 59, 119333 Moscow, Russia

**Keywords:** magnetron sputtering, coating, ZrB_2_, MoSi_2_, Zr-Mo-Si-B-(N), structure, mechanical and tribological properties, oxidation resistance

## Abstract

Coatings in a Zr-Mo-Si-B-N system were deposited by the magnetron sputtering of ZrB_2_-MoSi_2_ targets in argon and nitrogen. The structure of the coatings was investigated using scanning electron microscopy, X-ray diffraction, energy-dispersive spectroscopy, and glow-discharge optical emission spectroscopy. Mechanical and tribological properties were measured using nanoindentation and pin-on-disc testing. Oxidation resistance and oxidation kinetics were estimated via annealing in air at 1000–1500 °C and precision weight measurements. We found that the coatings deposited in Ar demonstrate a superior combination of properties, including hardness of 36 GPa, elastic recovery of 84%, a friction coefficient of 0.6, and oxidation resistance at temperatures up to 1200 °C. High oxidation resistance is realized due to the formation of the protective (SiO_2_ + ZrO_2_)/SiO_2_ oxide layer, which inhibits the diffusion of oxygen into the coating.

## 1. Introduction

Zirconium diboride is promising in multipurpose protective coatings for high-temperature units in aviation and space technology, the power engineering industry, metal-formed tools, cutting and machining tools, molds, etc., due to its extremely high melting temperature (3245 °C), high thermal conductivity (57.9 Wt·m^−1^·K^−1^), low coefficient of thermal expansion (5.9 × 10^−6^ °C^−1^) [1], high oxidation resistance (up to 2000 °C) [2], high hardness (22 GPa), and wear resistance [3]. The oxidation resistance of protective coatings can be enhanced via alloying by Si [4,5,6] or the introduction of silicon-containing phases, such as SiC, SiBC, and TaSi_2_ [7,8,9]. Silicon improves the oxidation resistance due to the formation of borosilicate glass on the surface of the material upon oxidation. This glass layer efficiently retards the diffusion of oxygen [10,11]. One of the most widely used high-temperature coatings is ZrB_2_-SiC [12,13,14], which demonstrates high protective properties and oxidation resistance up to 1500 °C. Over the last five years, there has been a considerable increase of interest toward ZrB_2_-MoSi_2_ coatings for the protection of carbon–carbon (C/C) and carbon–ceramic (C/SiC) composites from oxidation [15,16,17,18,19]. A comparison of ZrB_2_-SiC and ZrB_2_-MoSi_2_ coatings demonstrated that a MoSi_2_-alloyed specimen possessed a denser structure and higher oxidation resistance [20]. References [21,22] analyzed the effect of the introduction of MoSi_2_ and TaSi_2_ in ZrB_2_-based coatings. The ZrB_2_-MoSi_2_ coating experienced almost no oxidation at 1500 °C due to the formation of a dense oxide layer, whereas the ZrB_2_-TaSi_2_ coating formed a porous oxide layer that offered no protection against the influx of oxygen. In parallel, several dense ceramics and coatings are being developed based on MoSiB alloyed by various elements (Y, Al, Fe, Zr) [23,24,25,26].

ZrB_2_-MoSi_2_ and Mo-Si-B-Zr coatings are usually deposited using diffusion saturation [26,27], plasma deposition [21,28], flame spraying [19], and impregnation [29]. PVD and CVD are arguably the most interesting methods for the deposition of ZrB_2_-MoSi_2_ coatings from both a scientific and technological point of view, yet no such studies can be currently found in the literature. The closest analogs are coatings Zr-Si-B and Mo-Si-B, which were previously deposited by magnetron sputtering [5,8,25,30,31,32]. It should be noted that the properties of the coatings (in particular, tribological performance) can be greatly enhanced by the introduction of nitrogen into the coating via the sputtering in the nitrogen atmosphere [33,34].

This work aimed to deposit Zr-Mo-Si-B-(N) coatings via magnetron sputtering and investigate their structure, oxidation resistance, and mechanical and tribological properties.

## 2. Materials and Methods

Sputtering targets with a diameter of 120 mm and thickness of 5 mm were produced by combustion synthesis. The phase composition of the targets was investigated by XRD using a DRON-4 installation (RPE “Burevestnik”, St. Petersburg, Russia). A scanning electron microscope (SEM) S3400 (Hitachi, Tokyo, Japan) with an EDS add-on Noran 7 Thermo was used for the microstructural investigations. XRD analysis revealed that the ceramic target contained 89.5% h-ZrB_2_ with lattice parameters a = 0.3155 nm, c = 0.3507 nm, and 10.5% t-MoSi_2_ (a = 0.3209 nm, c = 0.7864 nm). Structural components could be easily recognized on the EDS maps, whereas the SEM images did not reveal clear phase boundaries (Figure 1).

The zirconium diboride matrix contained 1–5 µm inclusions of molybdenum silicide. The ceramic target was characterized by hardness of 19 GPa and fracture toughness of 6.69 MPa × m^1/2^.

The magnetron sputtering of the ceramic targets was performed in a direct current mode using a UVN-2M apparatus, described in detail elsewhere [35]. This apparatus employs a Pinnacle Plus 5 × 5 power supply (Advanced Energy) equipped with an arc-extinguishing system. The current and voltages were 2 A and 400–500 V, and the power was 1 kW. The target sputtering was performed in pure Ar (99.9995%) or N_2_ (99.999%) at a pressure of 0.1–0.2 Pa. The temperature of the substrate during deposition was ~350 °C. The bias voltage was not applied to the substrate. The substrate-target distance was about 80 mm. Plates made of VOK-100-1-grade alumina were used as the substrates. Before deposition, the substrates were cleaned by Ar^+^ ions with a 1.5–2 keV energy for 20 min at a pressure of 0.08 Pa using a slit-type ion source. The deposition duration was 40 min. To test the oxidation resistance of the coatings, they were annealed at 1000 °C for 1 h in the air using a SNOL-7.2/1200 muffle furnace. The elemental profiles and chemical composition of the coatings were analyzed using glow-discharge optical emission spectroscope (GDOES) PROFILER-2 (Horiba Jobin Yvon, Longjumeau, France) [36]. The set of standards from Horiba JY and additional bulk and film standards including TiN, TiCaCOPN, ZrN, TiB, and CrN were used for calibration. The cross-section fractures of the coatings were investigated using SEM Hitachi S4800 in a secondary electron emission regime at the accelerating voltage of 20–30 kV. The XRD analysis of the as-deposited and annealed coatings was performed on a Bruker D2 Phaser apparatus using the Bragg–Brentano configuration and CuKα radiation. Mechanical properties of the coatings were tested using the precision Nano-Hardness Tester (CSM Instruments, Peseux, Switzerland). The penetration depth at a 4 mN load did not exceed 10% of the coating’s thickness. The tribological measurements were performed at a Tribometer automated friction machine (CSM Instruments) using a pin-on-disc scheme with an alumina ball counterbody (diameter of 6 mm) at normal loads of 1 and 5 N. The wear tracks were analyzed using a Wyko-1100NT optical profilometer (Plainview, NY, USA) which was described in detail elsewhere [37].

## 3. Results and Discussion

According to the GDOES data, all the elements were distributed homogeneously across the coating’s thickness (Figure 2).

Depth-averaged elemental concentrations are provided in Table 1.

Coating 1A deposited in argon was characterized by a high concentration of boron (55 at.%) and zirconium (37 at.%), suggesting that the zirconium diboride was the main structural constituent of the coating. Additionally, the coating contained 4 at.% of Si and Mo. Coating 2N, deposited in N_2_, contained Zr, B, and N in approximately equal proportions (~30 at.%), 3 at.% Mo and 5 at.% Si. The content of admixtures was relatively low at 1.2 at.% O and 1.7 at.% C for the coating deposited in nitrogen and 0.04 at.% O and 0.22 at.% C for the coating deposited in Ar.

The chemical composition of the coatings was also measured using EDS in conjunction with the SEM investigation of the coating’s fractures. It should be noted that the EDS data on the content of light elements (especially boron) were likely erroneous. EDS data for Coating 1A suggests the 30–31 at.% boron content, which is more than 1.5 times below the GDOES data. For Coating 2N, the EDS data for boron content were 0–8 at.%, whereas the nitrogen content was as high as 70 at.%, which is unlikely for the condensed materials. Therefore, the GDOES data, in this case, is more credible.

SEM images of the cross-section fractures of the Coatings 1A and 2N are provided in Figure 3.

Both coatings possess a dense structure with no structural defects or columnar grains, which are characteristic of PVD coatings [38,39,40]. The fracture of Coating 1A conformed to the fracture of the substrate. The fracture of Coating 2N was characteristic of the amorphous coatings [41]. The thickness of Coatings 1A and 2N was estimated from the cross-section fracture images as 5.0 and 3.7 µm, correspondingly.

The growth rate of the SEM-derived coating was 125 and 93.5 nm/min for Coatings 1A and 2N, respectively. The decrease in the coating growth rate upon the transition from argon to nitrogen sputtering atmosphere can be explained by the easier ionization of argon as compared to nitrogen [42,43]. Additionally, the sputtered atoms from the target experience more scattering during their flight to the substrate when a heavier working gas is used. The measurement of the surface roughness via optical profilometry indicated that Coating 1 has the roughness of Ra = 68 nm, whereas the introduction of nitrogen reduced the surface roughness to Ra = 12 nm, which is close to Ra = 10 nm for the alumina substrate. The decrease in surface roughness with the introduction of nitrogen can be explained by the structural refinement of the coating [44,45].

The XRD patterns of the coatings were recorded at 2Ѳ = 10–120° diapason. Figure 4 contains a pattern fragment 2Ѳ = 20–70°, in which the phase reflexes with the highest intensity are located.

Samples 1A and 2N demonstrated peaks corresponding to the Al_2_O_3_ substrate (JCPDS 10-0173). Coating 1A deposited in Ar contained reflexes (001), (002), (112), and (210) of the hexagonal phase h-ZrB_2_ (JCPDS 89-3930). The Scherrer-equation-derived size of h-ZrB_2_ was 44–48 nm. The lattice parameters of h-ZrB_2_ were a = 0.320 nm and c = 0.345 nm, which are slightly different from the JCPDS card (a = 0.3165 nm, c = 0.3530 nm). The change in the lattice parameter might be related to the deviation of the h-ZrB_2_ phase from the stoichiometry. In the case of Coating 2N deposited in N_2_, the XRD pattern contained a broad reflex at 2Ѳ = 30–35°. No other reflexes corresponding to the coating were found. Therefore, the introduction of nitrogen resulted in the amorphization of the coating. Interestingly, the amorphous halo is located near the 2Ѳ value, which corresponds to the (111) reflex of the ZrN phase (JCPDS 89-5269). The increase in the nitrogen concentration resulted in the formation of the nitride bonds Zr-N, which were more energetically favorable and therefore hampered the formation of Zr-B bonds. This result supports the previous works [46,47] which demonstrated that in the Me-Si-B-C-N and Zr-B-C-N among all the possible bonds the Me-N (Me = Ti, Zr, Hf) are the most favorable. The formation of some B-N bonds in the 2N coatings is also possible. The location of the main peak of h-BN is close to 2Ѳ = 27° (JCPDS 85-1068).

In the case of 2N coatings, due to the competitive simultaneous nucleation and growth of the phases, the overall crystalline structure did not form, resulting in the formation of an amorphous coating. A similar effect was reported earlier for Cr-Al-Si-B-(N), Ti-B-Si-N, and Ti-B-C-N coatings [41,48,49].

Figure 5 provides the result of the TEM investigation of the produced coatings.

Coating 1A featured a columnar structure, with a 5–20 nm diameter of individual columnar grains. A similar microstructure was reported for the ZrB_2_ coatings, deposited on alumina substrate by magnetron sputtering [50]. One can see the conjunctions between the crystal lattices of adjacent grains in the microstructure (Figure 5a). Additionally, isolated amorphous areas are present between the crystallites. The measurements of the interplanar distance via HRTEM and SAED produced the following values: 0.3393, 0.2675, 0.2098, 0.1718, 0.1546, 0.1422, 0.1337, 0.1252, and 0.1143 nm, which correspond to the hexagonal h-ZrB_2_. Other elements in the coating (Mo and Si) form an amorphous phase. TEM investigation of Coating 2N revealed that its structure is amorphous, which agrees well with the XRD data.

The nanoindentation-derived mechanical properties of the coatings, including hardness, Young’s modulus, elastic recovery, and parameters H/E and H^3^/E^2^, are provided in Table 1. It should be noted that the H/E and H^3^/E^2^ ratios can be used for an assessment of the wear resistance, adhesion to the substrate, and fracture resistance of the coatings, as well as the type of localized deformation [51,52,53]. The non-reactive-deposited coating (Coating 1A) demonstrated higher hardness H = 36 GPa, Young’s modulus E = 415 GPa, elastic recovery W = 84%, and H^3^/E^2^ ratio = 0.271 GPa. The addition of N led to a decrease in mechanical properties; the hardness of Coating 2N was 14 GPa, E = 160 GPa, W = 64%. Both Coatings 1A and 2N were characterized by a H/E ratio = 0.09. A similar decrease in the mechanical properties was demonstrated in [54,55] at N content >15 at %, which may be associated with structural factors, in particular, with an increase in the concentration of the soft amorphous nitrogen-containing phase. The structural characteristics of the main phase determine the mechanical performance of the coatings. Therefore, the formation of hexagonal h-ZrB_2_ might affect the coating’s mechanical properties because the cubic c-ZrB is harder than h-ZrB_2_, but the latter has a higher resistance to elastic-plastic deformation [56].

For the first estimations, the tribological tests were conducted in relatively mild conditions: 1 N load, 50 m distance. For Coating 1A, the friction coefficient increased from 0.25 to 0.65 during the first 10 m distance and then stabilized around ~0.6 and remained constant until the end of the test. The nitrogen-containing coating (Coating 2N) had an initial coefficient of friction similar to Coating 1A. During the test, the coefficient of friction gradually increased and reached ~1.0 at a 50 m distance.

The analysis of the wear tracks revealed that Coating 1A successfully resisted wear, and the depth of penetration of the counterbody did not exceed 0.2 µm. Coating 2N experienced more pronounced wear, and the depth of wear track was 1.6 µm; however, the wear track depth did not exceed the coatings’ thickness of 3.7 µm. The wear rate of Coating 1A was hard to estimate due to the lack of pronounced wear. However, taking into account the surface roughness, we can conclude that the wear rate did not exceed 4.9 × 10^−6^ mm^3^N^−1^m^−1^. The profilometry-derived wear rate for Coating 2N was 9.0 × 10^−5^ mm^3^N^−1^m^−1^. Thus, Coating 1A demonstrated considerably higher wear resistance at a 1 N load and 50 m distance as compared to Coating 2N. A similar effect was reported in [55], where the introduction of the nitrogen into the magnetron-sputters ZrB_2_ coating resulted in a ~15% increase in the friction coefficient. Therefore, Coating 1A displayed a considerably higher wear resistance as compared to Coating 2N. Interestingly, both Coatings 1A and 2N were characterized by close values of the H/E parameters. Therefore, in our case, the wear resistance correlates better with the hardness and elastic resistance values, as well as with the H^3^/E^2^ ratio, which was higher for Coatings 1A (Table 1).

The next tribological characterization was conducted in harsher conditions: 5 N load and 1000 distance. The results of tribological tests are provided in Figure 6.

Coating 1A demonstrated a stable friction coefficient (0.80–0.88) at a 0–1000 m distance. The friction coefficient for Coating 2N increased from 0.14 to 0.83 at a 0–8.3 m distance. The low friction coefficient in the initial stage might be related to the presence of B-N bonds in the 2N coating. A similar effect was previously reported [34] in the investigation of the influence of nitrogen on the properties of boride coatings. After a short distance, the testing was stopped to fixate the moment of the complete wear of the coating.

The analysis of the wear tracks (Figure 7) revealed that Coating 1A successfully resisted wear, and the depth of penetration of the counterbody did not exceed 0.5 µm.

The wear rate of the 1A coating was 4.2 × 10^−7^ mm^3^N^−1^m^−1^. Importantly, the wear track depth of the nonreactive coating was 0.4 µm, which did not exceed the total coating’s thickness of 5 µm. Coating 2N experienced complete wear during the test, and the wear track depth was ~4 µm at the coating’s thickness of 3.7 µm.

Therefore, in our case, the wear resistance correlates better with the hardness and elastic resistance values, as well as with the H^3^/E^2^ ratio, which was higher for Coating 1A (Table 1). Notably, our previously investigated coatings ZrB_2_ and Zr-B-N demonstrated specific wear rates of 1.3 ÷ 8.2 × 10^−6^ mm^3^/(Nm), correspondingly [57], which is an order of magnitude higher than the wear rate of the 1A coating (Zr-Mo-Si-B). Specimen 1A was also characterized by higher wear resistance as compared to the baseline ZrB_2_ coating (2.8 × 10^−6^ mm^3^/(Nm)) [55].

Single-step isothermal annealing runs at 1000 °C with consecutive weight measurements revealed an increase in the mass of the first coating for 1.1 µg/cm^2^ after one run, whereas Coating 2N demonstrated a drop in mass 12 µg/cm^2^. The mass gain of Coating 1A is evidently caused by the formation of an oxide layer, whereas the mass decrease of Coating 2N might be related to the evaporation of the nonmetallic species (predominantly nitrogen) after reaction with oxygen. The SEM and EDS data of the oxidized coatings are presented in Figure 8 and Figure 9.

On the surface of Coating 1A, a three-layer oxide film formed, consisting of an upper 0.9 µm wide amorphous silica layer, followed by a lower layer consisting mostly of 450–700 nm zirconia crystallites in the silica matrix and the lowest layer of silica adjacent to the 3.2 µm thick nonoxidized part of the coating. Coating 2N after annealing in the same regime experienced complete oxidation and formed a single layer of polydisperse zirconia crystallites (bright spots in the microstructure) in the amorphous silica matrix (dark spots in the microstructure).

The XRD patterns of the Coatings 1A and 2N after annealing in air at 1000 °C are provided in Figure 10.

The annealing in air resulted in the formation of an oxide layer on the surface of Coating 1A. According to the XRD data, the layer was comprised of monoclinic m-MoO_2_ (JCPDS 76-1807), hexagonal h-MoO_3_ (JCPDS 65-0033), monoclinic m-ZrO_2_ (JCPDS 65-0728), and tetragonal t-ZrO_2_ (JCPDS 68-0200). Additionally, low-intensity peaks of the phase Al_4_Si_1_O_10_ (JCPDS 74-4144) were observed. This phase was formed as a result of interaction between the silicon and substrate on the boundary between substrate and coating. For Coating 1A, h-ZrB_2_ peaks were observed (JCPDS 89-3930) at 2Ѳ = 25.7, 32.6, 52.8, and 65.5°, signalizing the retention of the coating during the annealing. Coating 2N contained similar oxide phases m-MoO_2_, h-MoO_3_, m-ZrO_2_, t-ZrO_2_, and Al_4_Si_1_O_10_. The ZrN peak, which was present in the as-deposited coating, was not found after the annealing. The lack of phases characteristic for the as-deposited coating signifies complete oxidation of the 2N specimen, which is supported by the SEM investigation. The weight changes for Coatings 1A and 2N upon oxidation (Figure 11) were monitored in a step-wise manner after annealing runs in the air in the 200–1000 °C interval with 200 °C increments and dwelling time of 30 min. Up to a temperature of 600 °C inclusive, the behavior of Coatings 1A and 2N does not differ significantly. There is a slight decrease in the mass of the samples. The mass losses derived by approximation of the curves were 00013 and 00016 mg/min for the Coatings 1A and 2N, correspondingly. In the case of Sample 1A, the sharp increase in mass at temperatures of 600–1000 °C is probably due to the formation of SiO_2_ and ZrO_2_ oxide layers. The mass gain rate of the 1A sample in this temperature interval was 0.0067 mg/min.

On the other hand, the mass loss for sample 2N is due to the high losses of nitrogen atoms due to the volatilization of nitrogen oxide gas. In the 600–1000 °C, interval the mass decrease rate of the nitrogen-containing coating increased considerably, up to 0.004 mg/min. In comparison, the oxidation rate of Zr-Si-B coatings in similar conditions was ~0.01 mg/min [32].

To ascertain the peak oxidation resistance of the developed coatings, additional oxidation runs were performed using 10 min non-isothermal dwelling times (heating and cooling with a Nabertherm LHT 01/17 furnace at 10°/min) at 1300, 1400 and 1500 °C revealed that both Coatings 1A and 2N after annealing become optically transparent due to the complete oxidation and formation of a continuous oxide layer on top of the alumina substrate. Coating 1A was able to withstand a short-term isothermal oxidation run (10 min, 1200 °C, SNOL furnace) without complete oxidation. The results of the investigation of Coating 1A after this run are provided in Figure 12.

One can see that the coating contains a nonoxidized layer. The oxide layer formed at 1200 °C is structurally similar to the oxide layers formed at 1000 °C.

The baseline coating ZrB_2_ demonstrated low oxidation resistance: complete oxidation occurred upon heating in the air up to 700 °C for 1 h [8]. Coating 1A, deposited in this work, retained its protective properties up to 1200 °C. Therefore, the introduction of MoSi_2_ into the coating resulted in a massive increase in the oxidation resistance without any notable deterioration of the mechanical performance. The hardness of the ZrB_2_ coating and 1A specimen (Zr-Mo-Si-B) produced and investigated in similar conditions was 36–37 GPa. In comparison, our previously investigated ternary coatings Zr-Si-B and Mo-Si-B displayed higher oxidation resistance up to 1500–1700 °C [8,26], but their mechanical properties were more modest (H = 22–27 GPa).

## 4. Conclusions

Zr-Mo-Si-B coatings were deposited by magnetron sputtering in Ar and N_2_ using direct current. Two types of coatings with a thickness of 3.7–5.0 µm were deposited. The coatings deposited in Ar were mainly composed of hexagonal h-ZrB_2_ with crystallite sizes below 50 nm, whereas the coatings deposited in nitrogen were X-ray amorphous. The coatings deposited in Ar demonstrated superior mechanical, tribological, and chemical properties, with a hardness of 36 GPa, elastic recovery of 84%, H^3^/E^2^ ratio = 0.271 GPa, wear rate of 4.2 × 10^−7^ mm^3^N^−1^m^−1^, and oxidation resistance up to 1200 °C. The notable oxidation resistance of the Zr-Mo-Si-B coatings results from the formation of a protective surface layer consisting of an amorphous silica matrix with dispersed zirconia nanocrystallites (100–350 nm), which can efficiently impede the diffusion of oxygen into the unreacted coating.

## Figures and Tables

**Figure 1 materials-14-01932-f001:**
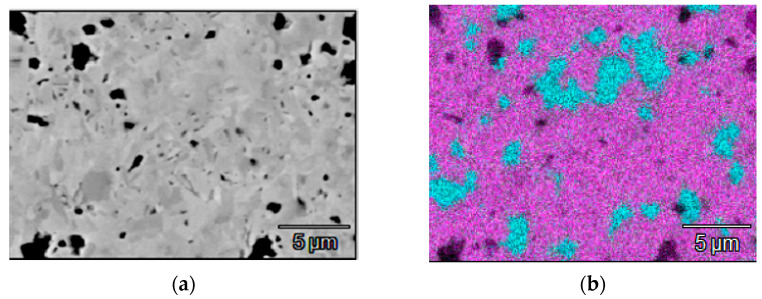
SEM image (**a**) and EDS map (**b**) of the cathode target (violet—Zr, blue—Si).

**Figure 2 materials-14-01932-f002:**
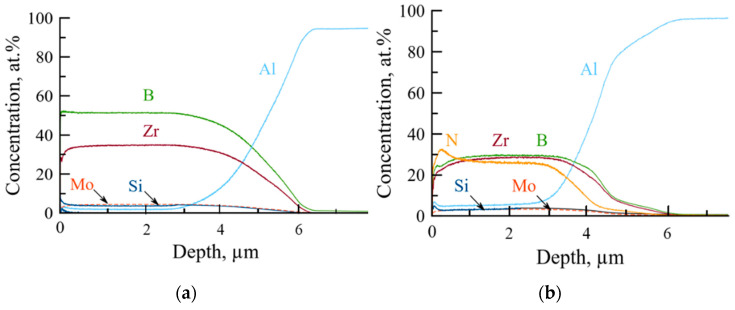
Elemental profiles of Coatings 1A (**a**) and 2N (**b**).

**Figure 3 materials-14-01932-f003:**
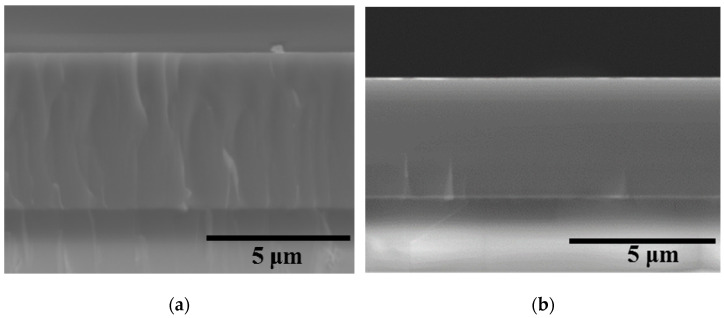
SEM images of Coatings 1A (**a**) and 2N (**b**).

**Figure 4 materials-14-01932-f004:**
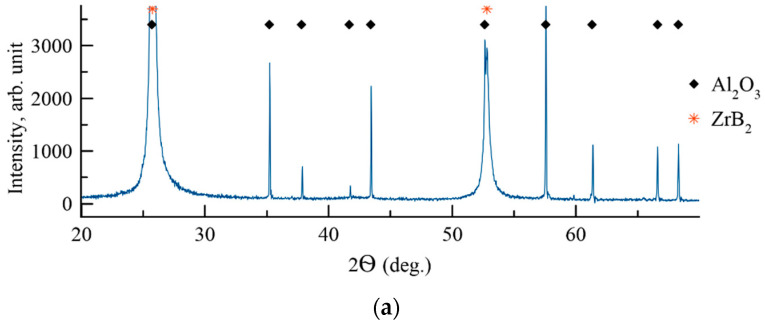
XRD patterns of Coatings 1A (**a**) and 2N (**b**).

**Figure 5 materials-14-01932-f005:**
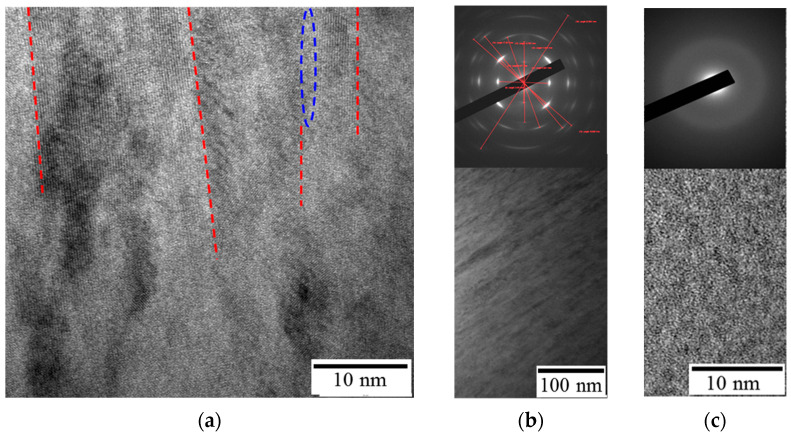
Cross-section TEM images and SAED patterns for Coatings 1A (**a**,**b**) and 2N (**c**). The red dashed lines correspond to the boundaries between columnar grains, and the blue oval shows amorphous region.

**Figure 6 materials-14-01932-f006:**
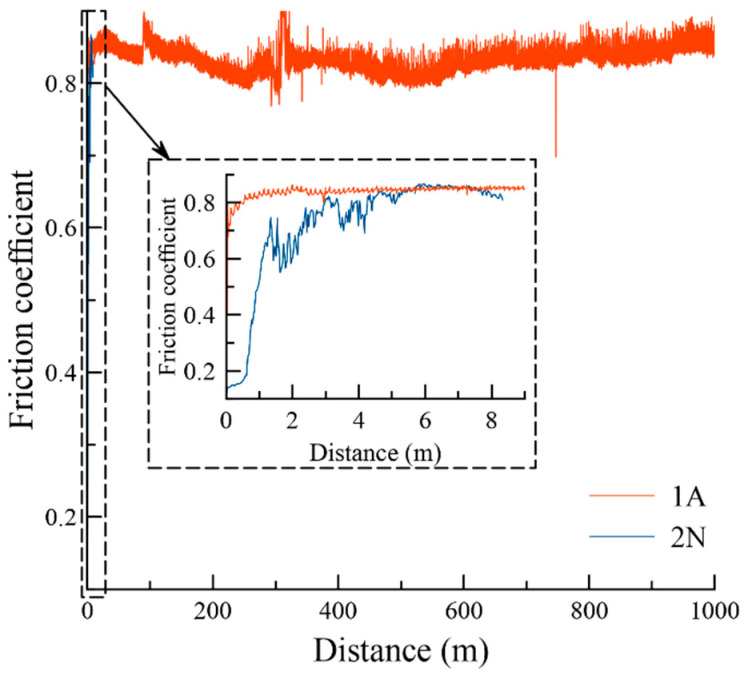
The friction coefficient for Coatings 1A and 2N.

**Figure 7 materials-14-01932-f007:**
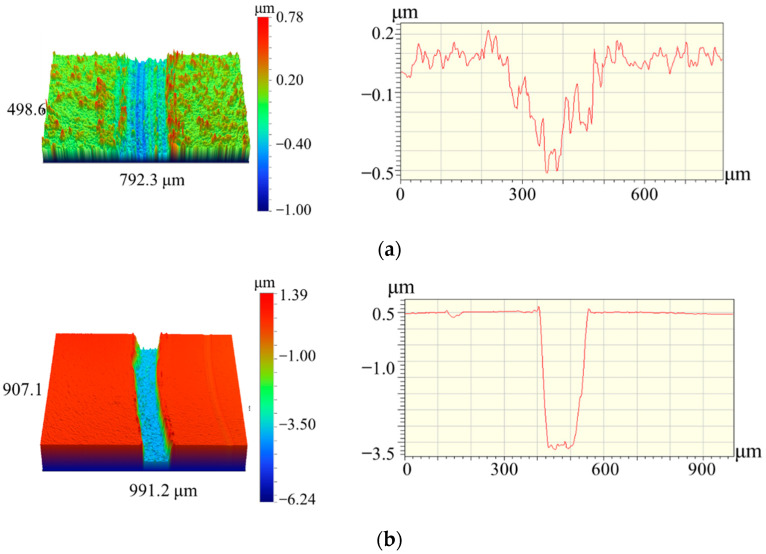
Three-dimensional (3D) and 2D images of the wear tracks for Coatings 1 (**a**) and 2 (**b**).

**Figure 8 materials-14-01932-f008:**
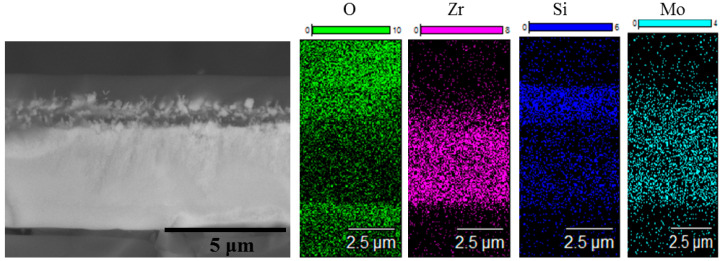
Cross-section SEM images of the Zr-Mo-Si-B coating (Coating 1A) after air annealing at 1000 °C for 1 h. The figure illustrates protection from oxidation.

**Figure 9 materials-14-01932-f009:**
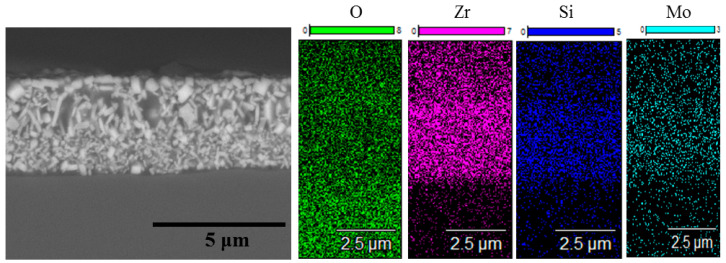
Cross-section SEM images of the Zr-Mo-Si-B-N coating (Coating 2N) after air annealing at 1000 °C for 1 h. The figure illustrates complete oxidation.

**Figure 10 materials-14-01932-f010:**
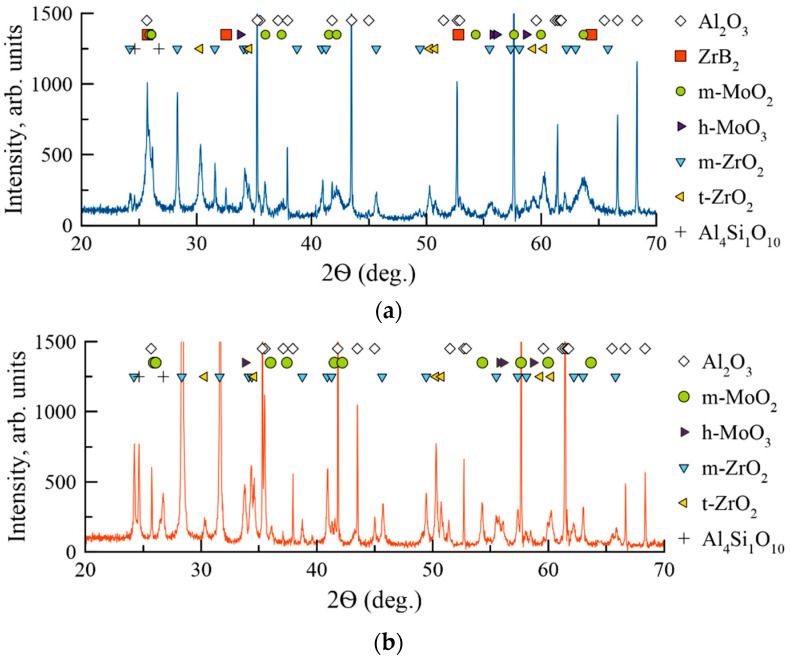
XRD patterns of Coatings 1A (**a**) and 2N (**b**) after annealing in air at 1000 °C.

**Figure 11 materials-14-01932-f011:**
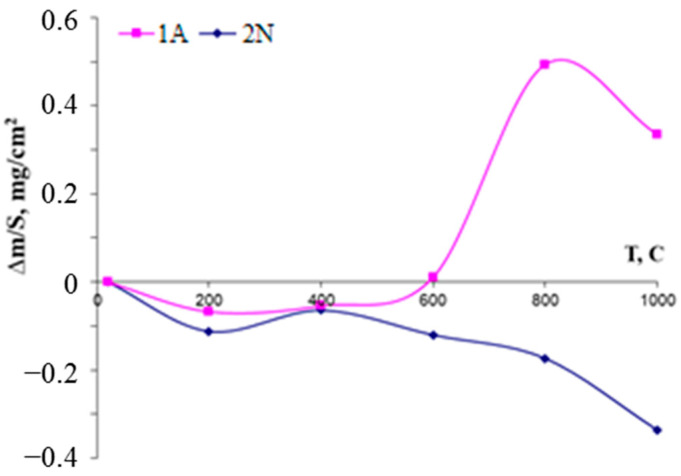
Weight changes of Coatings 1A and 2N in relation to the annealing temperature.

**Figure 12 materials-14-01932-f012:**
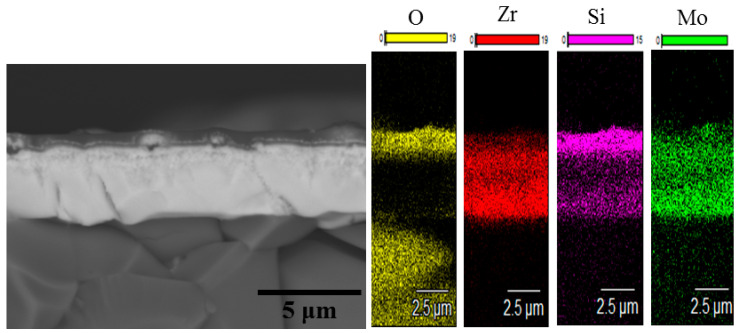
SEM data for the Zr-Mo-Si-B coating after exposition at 1200 °C for 10 min.

**Table 1 materials-14-01932-t001:** Composition and mechanical properties of the coatings.

№	Deposition Medium	Composition, at.%					H, GPa	E, GPa	H/E	H^3^/E^2^, GPa	W, %	Vw, Mm^3^/(Nm)
		Zr	Mo	Si	B	N						
1A	Ar	37	4	4	55	0	36	415	0.087	0.271	84	4.2 × 10^−^^7^
2N	N_2_	31	3	5	32	29	14	160	0.088	0.107	64	-

## Data Availability

The data presented in this study are available on request from the corresponding author.

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
