# Peer review of "Structure and Properties of Zr-Mo-Si-B-(N) Hard Coatings Obtained by d.c. Magnetron Sputtering of ZrB2-MoSi2 Target"

_materials, 2021, doi:10.3390/ma14081932_

Round 1
Reviewer 1 Report
The manuscript is devoted to the comparison of structure, composition mechanical, tribological properties and oxidation resistance of ZrMoSi-B and ZrMoSi-B-N coatings produced by DC magnetron sputtering in standard Ar and rather unusual N2 atmosphere, respectively. The work is based on an extensive set of standard experimental methods; the preparation of composite target by the authors should be emphasized. The idea of the work - combination of hard high temperature boride phase with silicon and molybdenum compounds providing better oxidation and tribological properties is quite interesting and improvements compared to standard materials of this class can be expected. However, these expectations were not fulfilled. First of all, only two coatings were involved in the study - one produced in Ar, the second in N2. Each sputtering, especially reactive sputtering, requires optimization of deposition parameters, even a feedback control to keep optimized conditions in the later case. It is not clear if the studied coatings were produced under optimized conditions. Without that, the comparison of their properties may not be fair. Moreover, comparison of one sample vs. one sample is statistically not sufficient. Both coatings exhibited high coefficients of friction and the introduction of nitrogen even dramatically reduced wear and oxidation resistance. Both studied systems were stable only up to 600oC which is also not remarkable. The main novelty, the application of nitrogen, caused substantial degradation of all properties. This casts a doubt on the eligibility of the approach and benefit from the publication of the work for scientific community.
The Results and discussion part, in fact, describes only the result of experimental observations and real discussion is missing. Few basic ideas in single sentences added in to the description of the data cannot substitute for their proper analysis .
Besides that, there are numerous smaller technical and formal issues which should be addressed.
1. the preparation and structure of the target (Fig. 1), though appreciated, does not relate to the main results of the work and should be a part of Materials and Methods.
2. deposition conditions are described very briefly; the mode of sputtering is missing, it can be found out only from the title of the work.
3. the way calibration of the signal intensity vs concentration in GDOES measurements was not specified.
4. in Results, it seems to be logical to start with the description of coating structure, thickness .... and not by composition from GDOES and EDS (which was found out to be limited and therefore, should not be considered...it EDS necessary in that case?) followed by structure and then back to composition from XRD and again back to coating structure by HRTEM.
5. What is the reliability of hardness measurements? Are you sure that the obtained values would be the same at the loads e.g. 3 mN or 4.5 mN? It is well known that 10% rule is necessary but definitely not sufficient for the determination of true mechanical properties of thin coatings. It should be used only as as an rule of thumb, i.e. very rough estimate with limited accuracy, and hardness depth profiles are required to determine indentation modulus and hardness according to the standards.
6. Friction tests were terminated after 50 m without well defined steady friction stage. Thus, it is difficult to estimate how representative these values are for the comparison of the behavior of both coatings. Moreover, friction alvays exhibits statistical variations and one experiments is not enough to obtain reliable value of the coefficient of friction.
7. Wear rate in ZrMoSiB-N was so low (around half of the coating thickness within 50 m) that it casts doubts on proper conditions (load) of the friction tests. The coating which would survive only 100 m sliding under that stress is useless.
8. double description of oxidation experiments (first prior to Fig. 8 and second in Fig. 10) is somehow confusing.
9. TG/DTA would provide much more accurate data than those in Fig. 10.
10. there seems to be a discrepancy between the data on thicknesses of different sublayers in the oxides cross section in the text and those from the scale in Fig. 8.
11. the preparation of the target should not be mentioned among the main conclusions.
Additional formal remarks include:
- the abbreviation of coating 1 and coating 2 in the text and marking them Ar and N2 in the plots is confusing and should be unified.
- The last paragraph prior to Conclusions contains two sentences from the instruction manual for the preparation of manuscript which should be deleted.
Based on doubts on real benefit of the work, statistical significance and even accuracy of some of the obtained results and on the amount of revisions required for its improvement, the work cannot be recommended for publication.
Author Response
Reviewer 1
- The manuscript is devoted to the comparison of structure, composition mechanical, tribological properties and oxidation resistance of Zr-Mo-Si-B and Zr-Mo-Si-B-N coatings produced by DC magnetron sputtering in standard Ar and rather unusual N2 atmosphere, respectively. The work is based on an extensive set of standard experimental methods; the preparation of composite target by the authors should be emphasized. The idea of the work - combination of hard high temperature boride phase with silicon and molybdenum compounds providing better oxidation and tribological properties is quite interesting and improvements compared to standard materials of this class can be expected. However, these expectations were not fulfilled.
Answer:
Indeed, this research was centered on the combination of the advantages of hard borides and silicon-based compounds. In this case, the baseline coating was ZrB2, which we investigated earlier and determined its low oxidation resistance: heating in the air up to 700 °С during 1 h resulted in a complete oxidation of the film [1]. In the current work, the ZrB2-MoSi2 films demonstrated considerably higher oxidation resistance. The protective properties of the coating deposited in argon persisted up to 1000°С (1 h dwelling) and 1200°С (10 min dwelling). Importantly, the hardness of the coatings did not decrease (36-37 GPa). Therefore, this work demonstrated the potential of the new coating composition and the positive effect of the alloying by silicon-based phase MoSi2.
- Kiryukhantsev-Korneev, F.V.; Lemesheva, M.V.; Shvyndina, N.V.; Levashov, E.A.; Potanin, A.Y. Structure, Mechanical Properties, and Oxidation Resistance of ZrB2, ZrSiB, and ZrSiB/SiBC Coatings. Prot. Met. Phys. Chem. Surfaces 2018, 54, 1147–1156, doi:10.1134/S207020511806014X.
- First of all, only two coatings were involved in the study - one produced in Ar, the second in N2. Each sputtering, especially reactive sputtering, requires optimization of deposition parameters, even a feedback control to keep optimized conditions in the later case. It is not clear if the studied coatings were produced under optimized conditions. Without that, the comparison of their properties may not be fair. Moreover, comparison of one sample vs. one sample is statistically not sufficient.
Answer:
During this stage, we investigated the properties of coatings deposited in pure Ar and N2. During the next stage, we will investigate the influence of the partial pressure of nitrogen on the structure and properties of the coatings. A similar research design was used for the investigation of coatings in the Ta-Si-Zr-B-C-N system: the approbation of pure gasses [1] with a subsequent investigation of the influence of the partial pressure of particular gasses [2]. Therefore, the optimization of composition, including the investigation of the flow rates of reactive gas on the structure and properties of the coatings, will be the subject of a follow-up article. The aim of the current article was the rapid estimation of the prospects of novel compositions and directions for further optimization. In particular, we established that the introduction of nitrogen is not optimal from the point of view of the mechanical and tribological properties (as well as oxidation resistance), but has a certain potential regarding the retention of an amorphous structure at elevated temperatures. Also, based on this work one can see that the initial point for the optimization has to be the non-reactive coating, with the further introduction of minor amounts of nitrogen (Are/N2 proportion = 10/1, 10/2, 10/4). The optimization-related research is rational, and it will be done in a future dedicated article.
- Kiryukhantsev-Korneev, Ph.V.; Sytchenko, A.D.; Levashov, E.A.; Lobova, T.A. Mechanical Properties and Heat Resistance of Ta–Zr–Si–B–C–N Coatings Obtained upon the Magnetron Sputtering of the TaZrSiB Target in Ar, N2, and C2H4 Atmosphere. Russian Journal of Non-Ferrous Metals, 2020, 61 (6), 732–738.
- Kiryukhantsev-Korneev, Ph.V.; Sytchenko A.D.; and etc. Structure, Oxidation Resistance, Mechanical, and Tribological Properties of N- and C-Doped Ta-Zr-Si-B Hard Protective Coatings Obtained by Reactive D.C. Magnetron Sputtering of TaZrSiB Ceramic Cathode. Coatings 2020, 10(10), 946
- Both coatings exhibited high coefficients of friction and the introduction of nitrogen even dramatically reduced wear and oxidation resistance.
Answer:
Indeed, the introduction of N decreases the performance of the coatings. Including the mechanical and tribological properties, as well as oxidation resistance. However, it has some positive effects – in situ measurements (including TEM with in situ heating) showed that the nitrogen-doped coating retains its amorphous structure up to 1000°С (data will be included in the next article), which might be useful for a range of applications. The number of compositions that retain their amorphous structures at such temperatures is limited. We hope that our article dedicated to the study of thermal stability of these coatings will be published shortly and you will have the opportunity to examine them in detail. The article in question includes holistic investigations of the 1A and 2N coatings in the temperature interval 20-1000°С using in-situ TEM, SAED, FFT, XRD, Raman, XPS, DSC, FTIR, nanoindentation, stress measurements.
- Both studied systems were stable only up to 600oC which is also not remarkable. The main novelty, the application of nitrogen, caused substantial degradation of all properties. This casts a doubt on the eligibility of the approach and benefit from the publication of the work for scientific community.
Answer:
One has to separate the oxidation resistance and thermal stability. The heat resistance was investigated by annealing in the air, whereas the thermal stability is usually measured by annealing in the vacuum. In our case, the Δm/S curve (Figure 10) does not characterize the thermal stability of the coatings, but rather the kinetics of their oxidation. In the case of the non-reactive coating, the mass gain was related to the formation of protective oxide layers. The mass loss in the case of N2-doped coating was related to the nitrogen fugacity. The in situ TEM and XRD investigations included in another article revealed that the phase compositions in both coatings do not happen up to 1000 ˚С, and therefore both coatings have high thermal stability. The current article is dedicated to the screening and characterization of the coatings deposited in Ar and N2, as well as to establishing the structure and properties–related trends.
- The Results and discussion part, in fact, describes only the result of experimental observations and real discussion is missing. Few basic ideas in single sentences added in to the description of the data cannot substitute for their proper analysis .
Answer:
Thank you for the suggestion. We added more data and discussion in the «Results and Discussion » section.
Besides that, there are numerous smaller technical and formal issues which should be addressed.
- the preparation and structure of the target (Fig. 1), though appreciated, does not relate to the main results of the work and should be a part of Materials and Methods.
Answer:
Following your remark, we moved the data on the manufacturing and characterization of the target (Fig. 1) to the «Materials and Methods» section.
- deposition conditions are described very briefly; the mode of sputtering is missing, it can be found out only from the title of the work.
Answer:
Thank you for the remark. We changed the manuscript following your suggestions. Deposition regime-related description was added: «The magnetron sputtering of the ceramic targets was performed in a direct current mode using a UVN-2M apparatus, described in detail elsewhere [36]. This apparatus employs Pinnacle Plus 5x5 power supply (Advanced Energy) equipped with an arc-extinguishing system. The current and voltages were 2 А and 400-500 V, the power was 1 kW. The target sputtering was performed in pure Ar (99.9995%) or N2 (99.999%) at a pressure of 0.1-0.2 Pa. The temperature of the substrate during deposition was ~350 ° C. Bias voltage wasn't applied to the substrate»
- the way calibration of the signal intensity vs concentration in GDOES measurements was not specified.
Answer:
The GDOES spectrometer was calibrated using the standards supplied by the manufacturer, which included bulk specimens of steels and alloys, as well as TiN coatings (manufactured by Ceratizit). For additional calibration, we used in-house fabricated bulk hot-pressed and combustion-synthesized ZrN, TiB, as well as coatings CrN and TiCaCOPN deposited by magnetron sputtering. Before the calibration, we calculated the relative sputtering rate using the etching depth profiling of the standard specimens and baseline sample (steel А11). During the calibration, we measured the signal intensity as related to the product of the concentration and relative etching rate. This procedure is standard for this kind of spectrometer.
The proposition «The set of standards from Horiba JY and additional bulk and film standards including TiN, TiCaCOPN, ZrN, TiB, and CrN were used for calibration.» was added to the text.
- in Results, it seems to be logical to start with the description of coating structure, thickness .... and not by composition from GDOES and EDS (which was found out to be limited and therefore, should not be considered...it EDS necessary in that case?) followed by structure and then back to composition from XRD and again back to coating structure by HRTEM.
Answer:
One should separate the elemental and phase compositions of the materials. The investigation of the chemical (elementary) composition using GDOES at the first stage of the research allows one to predict the phase composition of the coatings, which is consequently corroborated by the structural investigation (XRD, TEM). In our opinion, this sequence is logically sound. Additionally, the thickness-wise elemental profiles are important for the understanding of the uniformity of the phase composition of the coating. The sequence «deposition parameters-composition-structure-properties» is often used in coatings-related articles [1-5].
- Liu, X.; Iamvasant, C.; Liu, C.; Matthews, A.; Leyland, A. CrCuAgN PVD nanocomposite coatings: Effects of annealing on coating morphology and nanostructure, Applied Surface Science, 2017, 392, 732-746.
- Shaginyan, L.R.; Mišina, M.; Zemek, J.; Musil, J.; Regent, F.; Britun, V.F. Composition, structure, microhardness and residual stress of W–Ti–N films deposited by reactive magnetron sputtering, Thin Solid Films, 2002, 408, 136-147.
- Voevodin, A.A.; Hu, J.J.; Fitz, T.A.; Zabinski, J.S. Tribological properties of adaptive nanocomposite coatings made of yttria stabilized zirconia and gold, Surface and Coatings Technology, 2001, 146–147, 351-356.
- Baker, M.A.; Klose, S.; Rebholz, C.; Leyland, A.; Matthews, A. Evaluating the microstructure and performance of nanocomposite PVD TiAlBN coatings, Surface and Coatings Technology, 2002, 151–152, 338-343.
- Saringer, C.; Kickinger, C.; Munnik, F.; Mitterer, C.; Schalk, N.; Tkadletz, M. Thermal expansion of magnetron sputtered TiCxN1-x coatings studied by high-temperature X-ray diffraction, Thin Solid Films, 2019, 688, 137307.
- What is the reliability of hardness measurements? Are you sure that the obtained values would be the same at the loads e.g. 3 mN or 4.5 mN? It is well known that 10% rule is necessary but definitely not sufficient for the determination of true mechanical properties of thin coatings. It should be used only as as an rule of thumb, i.e. very rough estimate with limited accuracy, and hardness depth profiles are required to determine indentation modulus and hardness according to the standards.
Answer:
Our experiments were optimized to not penetrate the coatings for more than 10% of their thickness [1]. The thickness of 1A and 2N coatings was ~5 and 3 microns correspondingly, whereas the penetration depth was ~60 and ~100 nm. The dependence of the hardness on the penetration depth is a complicated one. It depends on the roughness of the surface, substrate properties, presence of residual stresses, etc. [2]. Investigation of the whole host of possible dependencies of that type is a massive and independent undertaking. For the different compositions of the coatings, the hardness vs indenter load curves can differ significantly [3]. It was previously demonstrated that the 4 mN load is optimal from the point of view of minimization of the influence of substrate and coating’s surface roughness [2, fig.4]. The decrease of the hardness values is usually observed at the depth below 60-100 nm [3,4]. This is supported by data from other authors [5,6].
- ISO 14577-1-4:2002. Metallic materials. Instrumented Indentation Test for Hardness and Materials Parameters.
- Xiao, Y.; Wu, L.; Luo, J.; Zhou, L. Mechanical response of thin hard coatings under indentation considering rough surface and residual stress, Diamond and Related Materials, 2020, 108, 107991
- Kiryukhantsev-Korneev, Ph.V.; Sheveiko, A.N. Specificity of Measurements of the Hardness of Thin Functional Coatings Using Sclerometry, Micro- and Nanoindentation Methods. Protection of Metals and Physical Chemistry of Surfaces, 2018, 54, 963–968.
- Levashov, E.A.; Shtansky, D. V.; Kiryukhantsev-Korneev, P. V.; Petrzhik, M.I.; Tyurina, M.Y.; Sheveiko, A.N. Multifunctional nanostructured coatings: Formation, structure, and the uniformity of measuring their mechanical and tribological properties. Russ. Metall. 2010, 2010, 917–935.
- Lofaj, F.; Németh, D. The effects of tip sharpness and coating thickness on nanoindentation measurements in hard coatings on softer substrates by FEM, Thin Solid Films, 2017, 644, 173-181.
- Bouzakis, K.-D.; Michailidis, N.; Skordaris, G. Hardness determination by means of a FEM-supported simulation of nanoindentation and applications in thin hard coatings, Surface and Coatings Technology, 2005, 200, 867-871.
- Friction tests were terminated after 50 m without well defined steady friction stage. Thus, it is difficult to estimate how representative these values are for the comparison of the behavior of both coatings. Moreover, friction always exhibits statistical variations, and one experiments is not enough to obtain reliable value of the coefficient of friction.
Answer:
Thanks to your suggestion, we were able to produce more informative and understandable data. We conducted additional testing using new conditions: 5 N load, 1000 m distance. Due to the increased distance, we were able to estimate the differences between the non-reactive and reactive coatings and evaluate the stability of the friction coefficients. The coatings deposited in Ar demonstrated a stable friction coefficient of ~0.85. The nitrogen-doped coating showed a friction coefficient of ~0.90 but was quickly worn out (during the first minutes of the test), resulting in the initial peak f>1. The results of tribological measurements were added to the manuscript.
- Wear rate in ZrMoSiB-N was so low (around half of the coating thickness within 50 m) that it casts doubts on proper conditions (load) of the friction tests. The coating which would survive only 100 m sliding under that stress is useless.
Answer:
Thanks to your comments, we were able to produce more convincing tribological data by increasing the load up to 5Н and distance up to 1000 m. It’s important to note that tribological testing was conducted in harsh conditions, using non-deformable alumina counter-body and substrate. Even in such taxing conditions, the non-reactive coating displayed a stable friction coefficient and high wear resistance of 4.2×10-7 mm3/(Nm) in the sliding wear mode. Additionally, one should note that the wear track depth of the non-reactive coating was 0.4 microns – way below the coating’s thickness of 5 microns. It should also be noted that the baseline coatings ZrB2 and Zr-B-N displayed a specific wear rate of 1.3-8.2×10-6 mm3/(Nm) correspondingly [1], which is an order of magnitude higher than the values for the Zr-Mo-Si-B coating. The wear resistance of the coating produced in the current work is also higher than that of ZrB2 coatings (2.8×10-6 mm3/(Nm)) investigated by Y. Dong et al. [2]. The nitrogen-doped coating experienced complete wear during the testing. Without a shadow of a doubt, the deposition in a 100% N2 atmosphere resulted in the decreased tribological performance. However, the data produced in the current work will serve as the baseline for the follow-up article, in which the coatings will be produced at Ar/N2 = 10/1, 10/2, 10/4.
- Kiryukhantsev-Korneev, Ph.V.; Sytchenko, A.D. The Influence of H, W, H/E, H3/E2, Structure and Chemical Composition on the Resistance of Ti–B–(N), Mo–B–(N), Cr–B–(N), and Zr–B–(N) Coatings to Cyclic Impact Loading. Protection of Metals and Physical Chemistry of Surfaces. 2020, 56(6), 1190-1200, doi:10.1134/S2070205120060143
- Dong, Y.; Wang, T.G.; Yan, B.; Qi, H.J.; Guo, Y.Y.; Xu, S.S. Study on the microstructure and mechanical properties of Zr-B-(N) tool coatings prepared by hybrid coating system. Procedia Manufacturing. 2018, 26, 806–817, doi: 10.1016/j.promfg.2018.07.099
- double description of oxidation experiments (first prior to Fig. 8 and second in Fig. 10) is somehow confusing.
Answer:
Our investigation included various kinds of annealing experiments:
- Long-term single-step annealing at 1000 ˚С with 60 min dwelling time (Fig.8 и 9). After the test, the specimens were weighted to estimate the mass gain, and were structurally characterized using SEM and EDS.
- Step-wise annealing in the air in the 200–1000 °С intervals with 200 °С steps and 30 min dwelling time (Fig. 10). After the annealing at Т=20, 200, 400, 600, 800 и 1000 °С the specimens were weighted to estimate the mass changes, oxidation kinetics, and thermal cycling resistance.
- Short-term annealing at 1200 С with 10 min dwelling time (Fig.11) was conducted only for the non-reactive coating since it displayed a higher oxidation resistance at 1000 ˚С.
Therefore, figures 8 and 10 provide the results of annealing experiments varied temperatures and dwelling times. To ensure more clarity, we added a short description of annealing conditions (temperature and dwelling time) before every related figure.
- TG/DTA would provide much more accurate data than those in Fig. 10.
Answer:
We agree with the remark. In the follow-up article dedicated to the investigation of the thermal stability of Zr-Mo-Si-B coatings, we plan to conduct the DSC experiments using the STA 449 F1 Jupiter installation (Netzsch). The coatings will be deposited on the alumina micro-crucibles and heated up to temperatures above 1000 ˚С in Ar and Ar+O2 environments.
- there seems to be a discrepancy between the data on thicknesses of different sublayers in the oxides cross section in the text and those from the scale in Fig. 8.
Answer:
We are grateful for your comment, which allowed us to increase the quality of our article. The following correction was made: «On the surface of the coating 1A, a 3-layer oxide film has formed, consisting of an upper 0.9 µm-wide amorphous silica layer, followed by a lower layer consisting mostly of 450-700 nm zirconia crystallites in silica matrix, and the lowest layer of silica adjacent to the 3.2 µm thick non-oxidized part of the coating.»
- the preparation of the target should not be mentioned among the main conclusions.
Answer:
According to your suggestion, the proposition «The ZrB2-10.5%MoSi2 composite target was manufactured by combustion synthesis and used for magnetron sputtering of coatings in Ar and N2 using a constant current.» was replaced by «Zr-Mo-Si-B coatings were deposition by magnetron sputtering in Ar and N2 using direct current.»
Additional formal remarks include:
- the abbreviation of coating 1 and coating 2 in the text and marking them Ar and N2 in the plots is confusing and should be unified.
Answer:
Following your comment, as well as comments from other Reviewers, we introduced in the manuscript notifications «1A» for the coating deposited in argon and «2N» for the coating deposited in nitrogen.
- The last paragraph prior to Conclusions contains two sentences from the instruction manual for the preparation of manuscript which should be deleted.
Answer:
We thank you for your comment, which allowed us to increase the quality of our article. The manuscript was corrected accordingly.
Reviewer 2 Report
The manuscript reported coatings in the Zr-Mo-Si-B-N system deposited by magnetron sputtering of ZrB2-MoSi2 targets in argon and nitrogen atmospheres, respectively. Literatures were reviewed in the Intro part. The two kinds of coatings were systematically characterized, which are interesting in the context of anti-oxidation coatings at high temperatures. The following are suggested for further revision:
- Materials and Methods
Ar+ ions, super script of +
- Results and discussion
Fig.2, The samples were not named in the main text and in figure captions. The names of C1Ar and C2N may contain deposition information, for convenient identification.
Table 1, In comparison of the compositions of 1 and 2, it seems N partially substitutes B in coating 2. Is there any difference in affinity between Zr-B and Zr-N bindings? Please note the data were obtained from GDOES.
Fig.4 and Fig.5, why coating 2 is amorphous? The mechanism could be further explored based on refs, e.g. amorphous Si-B-N phases.
Fig.5a, the text of part is not complete. Please specify planar or vertical view of the photos.
Fig.8, Fig.9, Fig.11, please indicate the sub-layers and the element photos with notations.
Fig.10, please note the curves with coatings 1 and 2. The figure needs improvement.
‘at 10 °/min) at 1300, 1400 и 1500 °С revealed’ N—and.
Author Response
Reviewer 2
The manuscript reported coatings in the Zr-Mo-Si-B-N system deposited by magnetron sputtering of ZrB2-MoSi2 targets in argon and nitrogen atmospheres, respectively. Literatures were reviewed in the Intro part. The two kinds of coatings were systematically characterized, which are interesting in the context of anti-oxidation coatings at high temperatures. The following are suggested for further revision:
- Materials and Methods
Ar+ ions, super script of +
Answer:
We are grateful for your comment, which allowed us to increase the quality of our article. The manuscript was corrected accordingly.
- Results and discussion
Fig.2, The samples were not named in the main text and in figure captions. The names of C1Ar and C2N may contain deposition information, for convenient identification.
Answer:
Following your proposition, we introduced into the manuscript the notifications «1A» for the coatings deposited in Ar and «2N» for the coatings deposited in nitrogen.
- Table 1, In comparison of the compositions of 1 and 2, it seems N partially substitutes B in coating 2. Is there any difference in affinity between Zr-B and Zr-N bindings? Please note the data were obtained from GDOES.
Answer:
Unlike XPS or Raman, the GDOES technique provides no data on the chemical bindings in the coating. However, we conducted the TEM investigation at room temperature and with in situ heating up to 1000 ˚С (the data will be published in the follow-up article). No ZrBx or ZrN phases were formed, and the coating retained its amorphous structure up to 1000 ˚С. In the follow-up article, we plan to conduct an XPS study for the as-deposited coatings and after annealing at temperatures up to 1000 ˚С.
- Fig.4 and Fig.5, why coating 2 is amorphous? The mechanism could be further explored based on refs, e.g. amorphous Si-B-N phases.
Answer:
Thank you for your comments, which allowed us to improve the manuscript.
It is known that the sputtering in nitrogen produces a nitride layer on the target, which has a considerably lower sputtering rate as compared to pure metals, which leads to a decrease in the coating’s growth rate and rise of the nitrogen content. An increase in nitrogen content results in the emergence of Zr-N bindings. The Gibbs free energy calculated considering the deposition parameters, for the Zr-B is ΔG = -694 kJ, for Zr-N is ΔG = -584 kJ and for В-N is ΔG = -584 kJ. Therefore, Zr-N and B-N bindings will be formed predominantly and will hamper the formation of Zr-B bindings. Thus, in the Me-Si-B-C-N and Zr-B-C-N systems, the Me-N bindings (Ме= Ti, Zr, Hf) are the most prominent [1-3].
According to the equilibrium ternary phase diagram (Fig.1), in the coating 2N, the formation of ZrN, BN, and ZrB2 is possible. As a result of the competing growth of the phases, the formation of the crystalline structure is inhibited and the coatings adopt an amorphous microstructure. An analogous effect was reported for the coatings Тi-B-Si-N, Zr–B–C–N, and Ti-B-C-N at N concentration above 20 at.% [4,5].
This discussion was added to the manuscript per your comment.
Fig.1. Ternary phase diagram (convex hull) of the Zr(Mo)-B(Si)-N system. The composition of 2N coating is marked by a black dot.
- Houska, J.; Mares, P.; Simova, V.; Zuzjakova, S.; Cerstvy, R.; Vlcek, J. Dependence of characteristics of MSiBCN (M=Ti, Zr, Hf) on the choice of metal element: Experimental and ab-initio study. Thin Solid Films. 2016, 616, 359-365.
- Houska, J.; Kohout, J.; Vlcek, J. Effect of N and Zr content on structure, electronic structure and properties of ZrBCN materials: An ab-initio study. Thin Solid Films. 3013, 542, 225-231.
- Vlček, J.; Steidl, P.; Kohout, J.; Čerstvý, R.; P.; Zeman, Prokšová Š.; Peřina V.;, Hard nanocrystalline Zr–B–C–N films with high electrical conductivity prepared by pulsed magnetron sputtering, Surface and Coatings Technology, 2013, 215, 186-191.
- Pleva, M.; Grančič, B.; Mikula, M.; Truchlý, M.; Roch, T.; Satrapinskyy, L.; Gregor, M.; Ďurina, P.; Girman, V.; Švec, P.; Plecenik, A.; Kúš, P. Thermal stability of amorphous Ti-B-Si-N coatings with variable Si/B concentration ratio. Surf. Coat. Technol. 2018, 333, 52-60.
Lin, J.; Moore, J.J.; Mishra, B.; Pinkas, M.; Sproul, W.D. The structure and mechanical and tribological properties of TiBCN nanocomposite coatings. Acta Materialia. 2010, 58 (5), 1554-1564.
- Fig.5a, the text of part is not complete. Please specify planar or vertical view of the photos.
Answer:
The ТEM images were acquired from the transverse cut of the samples. This information was added to the Fig.5 subscription: «TEM images (сross-section) and SAED patterns for coatings 1A (a, b) and 2N (c)…»
- Fig.8, Fig.9, Fig.11, please indicate the sub-layers and the element photos with notations.
Answer:
Thank you for the comment. The subscriptions which indicated elements were distorted due to the formatting. In the corrected version, each subscription is located right under the corresponding EDS map. No sub-layers were deposited under the Zr-Mo-Si-B coatings.
- Fig.10, please note the curves with coatings 1 and 2. The figure needs improvement.
‘at 10 °/min) at 1300, 1400 и 1500 °С revealed’ N—and.
Answer:
Thank you for the comment. We corrected Fig. 10 by adding notifications «1A» for the coatings deposited in Ar and «2N» for the ones deposited in nitrogen.
Reviewer 3 Report
I inform few improvements about this paper.
- It is better to add application examples of industrial products at the Introduction, like as " coatings for cutting tools ".
- It's need to add the details of deposition conditions, temperature , substrate bias voltages , pre-treatment conditions and any other. In general it's give the great influence to the properties of coatings.
- In usually, sputtering process to nitride films needs process gas of Ar/N2 mixed gas. If the process gas of coatings 2 is Ar/N2 mixed gas, Modify the description of P.2 L. 62 "~in Ar(99.9995%) or N2(9.999%) at~", and add the concentration ratio of the mixed gas.
- In usually the material of hexagonal crystal systems decrease density and hardness than cubic systems, TiAlN, AlCrN and any other hard coatings. I hope to add any comments about c-ZrB coatings. And I hope to keep the investigation about c-ZrBMoSi.
- I hope to add any comments about the film properties dependence of the concentrates of MoSi2.
Please consider his matter.
Author Response
Reviewer 3
I inform few improvements about this paper.
- It is better to add application examples of industrial products at the Introduction, like as " coatings for cutting tools ".
Answer:
Thank you for your insights, which helped us improve the paper and expand upon the field of application of the developed coatings. The proposition “Zirconium diboride is a perspective material for multi-purpose protective coatings due to its extremely high melting temperature (3245 °C), high thermal conductivity (57.9 Wt·m-1·K-1), low coefficient of thermal expansion (5.9×10-6 °С-1) [1]…» was corrected according to your comment: «Zirconium diboride is promising as a multi-purpose protective coating for high-temperature units of aviation and space technology, power engineering industry, metal formed tools, cutting and machining tool, molds, etc.»
- It's need to add the details of deposition conditions, temperature , substrate bias voltages , pre-treatment conditions and any other. In general it's give the great influence to the properties of coatings.
Answer:
We agree with your comment and, correspondingly, added a clarification to the description of the coating deposition method. No external heating was applied to the substrate. The substrate was heated up to 350 °С by magnetron sputtering process due to the kinetic energy of the atoms of deposited material, plasma radiation, as well as energy release due to the condensation and crystallization. Since alumina is a dielectric, no voltage bias was applied.
- In usually, sputtering process to nitride films needs process gas of Ar/N2 mixed gas. If the process gas of coatings 2 is Ar/N2 mixed gas, Modify the description of P.2 L. 62 "~in Ar(99.9995%) or N2(9.999%) at~", and add the concentration ratio of the mixed gas.
Answer:
In this work, the coatings were deposited in two atmospheres: 100% Ar and 100%N2. The clarifications were added to the «Methods» section. In a follow-up article, we plan to investigate the coatings deposited at different Ar/N2 rations - 10/1, 10/2, 10/4. This will allow us to establish the influence of the nitrogen content on the structure and properties of the coatings according to the algorithm previously applied to Ta-Si-Zr-B-C-N coatings [1].
- Kiryukhantsev-Korneev, Ph.V.; Sytchenko A.D.; et. al. Structure, Oxidation Resistance, Mechanical, and Tribological Properties of N- and C-Doped Ta-Zr-Si-B Hard Protective Coatings Obtained by Reactive D.C. Magnetron Sputtering of TaZrSiB Ceramic Cathode. Coatings 2020, 10(10), 946
- In usually the material of hexagonal crystal systems decrease density and hardness than cubic systems, TiAlN, AlCrN and any other hard coatings. I hope to add any comments about c-ZrB coatings. And I hope to keep the investigation about c-ZrBMoSi.
Answer:
We agree with the comment. It is known that the hardness of c-ZrB can surpass that of h-ZrB2 [1]. In our case, the atomic proportion of Zr:B was 1:2, and a hexagonal ZrB2 was formed. In future, we might undertake research dedicated to the optimization of coating’s composition by varying the boron and zirconium content to achieve Zr:B ratio of 1:1 and promote the formation of cubic c-ZrB phase. This new research direction is interesting; we are thankful for your recommendation and will use them in our future works.
- Li, H.; Zhang, L.; Zeng, Q.; Wang, J.; Cheng, L.; Ren, H.; Guan K. Crystal structure and elastic properties of ZrB compared with ZrB2: A first-principles study. Computational Materials Science. 2010, 49 (4), 814-819.
- I hope to add any comments about the film properties dependence of the concentrates of MoSi2.
Answer:
We constantly establish these relations in our research; in particular, we investigated coatings deposited using ceramic cathodes MoSi2 [1], ZrB2 [2,3], ZrB2+80%(90%MoSi2+10%MoB) [4], ZrB2+20%(90%MoSi2+10%MoB) (current work) and ZrB2+5%(90%MoSi2+10%MoB) (follow-up work in progress).
The coating ZrB2 was chosen as a baseline to investigate the influence of MoSi2 doping on the structure and properties of the coatings. Our previous investigations revealed that the coating ZrB2 suffered from a low oxidation resistance: annealing in the air at 700 °С and dwelling time of 1 hour has led to the complete oxidation of the coating [2]. In the current work, the coatings ZrB2-MoSi2 demonstrated a considerably higher oxidation resistance. The protective properties of the coating deposited in Ar were retained up to 1000 °С (1 h dwelling) and 1200 °С (10 min dwelling). The hardness of the coatings (36-37 GPa) did not diminish as compared to ZrB2. Therefore, this work demonstrates the positive impact of the MoSi2 alloying and the promise of the developed novel coating compositions. At the same time, the coatings with a high content of MoSi2 can withstand oxidation at extreme temperatures (1500-1700 °С [5]), but their hardness usually does not exceed 30 GPa, whereas the ZrB2 coatings display hardness of approx. 40 GPa [2].
- Kiryukhantsev-Korneev, Ph.V.; Potanin, A.Yu. Structure, Mechanical Properties, and Oxidation Resistance of MoSi2, MoSiB, and MoSiB/SiBC Coatings. Russian Journal of Non-Ferrous Metals, 2018, 59 (6), 698–708.
- Kiryukhantsev-Korneev, F.V.; Lemesheva, M.V.; Shvyndina, N.V.; Levashov, E.A.; Potanin, A.Y. Structure, Mechanical Properties, and Oxidation Resistance of ZrB2, ZrSiB, and ZrSiB/SiBC Coatings. Prot. Met. Phys. Chem. Surfaces 2018, 54, 1147–1156.
- Kiryukhantsev-Korneev, Ph.V.; Sytchenko, A.D. The Influence of H, W, H/E, H3/E2, Structure and Chemical Composition on the Resistance of Ti–B–(N), Mo–B–(N), Cr–B–(N), and Zr–B–(N) Coatings to Cyclic Impact Loading. Protection of Metals and Physical Chemistry of Surfaces. 2020, 56(6), 1190-1200.
- Kiryukhantsev-Korneev, Ph.V.; Loginov, P.A.; Orekhov, A.S.; Levashov, E.A. Study of nanomechanical properties of thin films using in-situ “Push-to-Pull” method in the column of transmission electron microscope Journal of Physics: Conference Series. 2020, 1688, 012004.
- Kiryukhantsev-Korneev, P.V.; Iatsyuk, I.V.; Shvindina, N.V.; Levashov, E.A.; Shtansky, D. V. Comparative investigation of structure, mechanical properties, and oxidation resistance of Mo-Si-B and Mo-Al-Si-B coatings. Corros. Sci. 2017, 123, 319–327.
Round 2
Reviewer 1 Report
The work compares structure, mechanical and tribological properties and oxidation behavior of Zr-B-(Mo-Si) and Zr-B-N -(Mo-Si) coatings deposited by DC magnetron sputtering by Ar and reactive DC magnetron sputtering by nitrogen from ZrB2+MoSi2 target. Boride-based coating was found to be textured, hard, stiff, exhibiting high wear resistance and reasonable oxidation resistance due to formation of thin and dense silica and zirconia scale. In contrary, boron nitride based coating was found to be amorphous, soft, with poor wear and oxidation resistance due to the release of nitrogen and formation of porous oxide scale. Despite the use of nitrogen for sputtering and as a reactive gas is rather unusual and may interesting possibilities, it was not successful and the studied boron nitride based coatings should not be used. Thus, the only positive result was identified from the comparison of the current ZrB2-MoSi2 with the reference pure ZrB2 in the literature, where and increase of oxidation resistance from 700oC to around 1000oC at comparable hardness (prior to oxidation) was emphasized. However, the significance and novelty of this result is questionable, especially if the authors claim that similar Zr-Si-B and Mo-Si-B systems resist up to 1500-1700oC. Some other issues should also be addressed:
- the whole work is based on 1 boride- and 1 boride nitride-based coating. It is not clear if the conditions for the deposition of each coating were optimized to have fair comparison. Obviously, the repeatability of the coating deposition, structure and properties cannot be tested and their statistical reliability is also not known. One experiment is definitely not enough.
- Coating 2N was completely worn out after 20 m but the experiment was run for 1000 m. It is obvious that the obtained friction curve relates to the friction between alumina ball and alumina substrate and the role of the coating was lost. The comparison of its friction curve with the curve from 1A is senseless.
- how wear resistance of the coating 2N was calculated - from the whole sliding distance or from the first 20 m (and thickness of coating)? The part related to tribological tests seems to be questionable and its removal and refocus of the work toward oxidation resistance is recommended.
- 4. the authors identified the differences in the oxide scales formed in both studied systems but their discussion was limited to the description of microscopic differences without any attempt of the analysis of physical model. Even the comparison of the porosity in these scales, which would provide the most simple explanation of the difference in oxidation resistance, was missing. The comparison of the weight gain due to oxidation and weight loss due to nitrogen release would be another simple estimation expected in the work on oxidation resistance.
The manuscript, despite describing interesting system, interesting techniques and appreciable amount of work involved, brings only very simple results without principal novelty, statistical reliability and without deeper understanding of the processes in friction and oxidation behavior. The work may be suitable for the conference proceeding but it cannot be recommended for the publication in a serious scientific journal.
Author Response
Reviewer #1
Comment 1: The work compares structure, mechanical and tribological properties and oxidation behavior of Zr-B-(Mo-Si) and Zr-B-N -(Mo-Si) coatings deposited by DC magnetron sputtering by Ar and reactive DC magnetron sputtering by nitrogen from ZrB2+MoSi2 target. Boride-based coating was found to be textured, hard, stiff, exhibiting high wear resistance and reasonable oxidation resistance due to formation of thin and dense silica and zirconia scale. In contrary, boron nitride based coating was found to be amorphous, soft, with poor wear and oxidation resistance due to the release of nitrogen and formation of porous oxide scale. Despite the use of nitrogen for sputtering and as a reactive gas is rather unusual and may interesting possibilities, it was not successful and the studied boron nitride based coatings should not be used. Thus, the only positive result was identified from the comparison of the current ZrB2-MoSi2 with the reference pure ZrB2 in the literature, where and increase of oxidation resistance from 700oC to around 1000oC at comparable hardness (prior to oxidation) was emphasized. However, the significance and novelty of this result is questionable, especially if the authors claim that similar Zr-Si-B and Mo-Si-B systems resist up to 1500-1700oC.
Author response :
Dear Reviewer, I want to emphasize that the Zr-Mo-Si-B coatings produced in this work are superior to the baseline ZrB2 coatings. Following the comment of other reviewer, we added a paragraph containing discussion of the influence of the MoSi2 alloying on the structure and properties of ZrB2 coatings. This paragraph references the previously investigated coatings Zr-Si-B and Mo-Si-B, which are resistant to oxidation up to 1500-1700oC. To be fair, however, we have to note that: a) those coatings had an increased Si content up to 50-60 at.%, so the matrix of the coatings was comprised of silicides rather than borides. The increase in the silicon content proportionally improves the oxidation resistance. In our work we investigate zirconium diboride-based coatings and ways to improve their oxidation resistance. b) mechanical properties of Mo-Si-B and Zr-Si-B coatings are substantially below those of Zr-Mo-Si-B. In particular, hardness is 10 GPa (or 30%) lower. Therefore, the results of our work are noteworthy in that regard.
It should be noted that the 2N coating is attractive due to its high thermal stsability. Below are the in-situ TEM data at different temperatures:
20C
800C
1000C
Amorhpous structure is retained up to 1000С, which shows a definite promise of this composition.
Comment 2: The whole work is based on 1 boride- and 1 boride nitride-based coating. It is not clear if the conditions for the deposition of each coating were optimized to have fair comparison. Obviously, the repeatability of the coating deposition, structure and properties cannot be tested and their statistical reliability is also not known. One experiment is definitely not enough.
Author response :
The second coating was not based on BN. The coating was amorphous, with Zr, B, and N concentrations close to equiatomic (30 at.%).
We do not agree with this comment. In this work, we deposited 10 samples of each composition. The measurements were reproduced enough time to collect a meaningful statistic. GDOES measurements of elemental compositions were performed in 3 points, SEM-EDS of fractures – from 3 to 5 points. Nanoindentations were performed at ≥ 9 points, etc. The step-wise variation of nitrogen content, including the pulsed and HIPIMS deposition, will be the topic of the next series of experiments and the follow-up articles.
Comment 3: Coating 2N was completely worn out after 20 m but the experiment was run for 1000 m. It is obvious that the obtained friction curve relates to the friction between alumina ball and alumina substrate and the role of the coating was lost. The comparison of its friction curve with the curve from 1A is senseless. how wear resistance of the coating 2N was calculated - from the whole sliding distance or from the first 20 m (and thickness of coating)? The part related to tribological tests seems to be questionable and its removal and refocus of the work toward oxidation resistance is recommended.
Author response :
We agree completely with this comment. The wear rate can be measured correctly only if the coating was not worn completely. Hence, the wear rate of the coating 2N at 1000 , distance cannot be measured correctly. The coating 2N, however, sustained testing at 2N load and a short distance of 50 m. This is enough for estimations of wear rate and comparison with coating 1A tested in the same conditions. Appropriate corrections were made in the text, table, and figures. We additionally discuss the tribological data produced in mild and harsh testing conditions.
Comment 4: The authors identified the differences in the oxide scales formed in both studied systems but their discussion was limited to the description of microscopic differences without any attempt of the analysis of physical model. Even the comparison of the porosity in these scales, which would provide the most simple explanation of the difference in oxidation resistance, was missing. The comparison of the weight gain due to oxidation and weight loss due to nitrogen release would be another simple estimation expected in the work on oxidation resistance.
Author response :
In this article we estimate the oxidation resistance using annealing in air with subsequent weighting and microscopic characterization. These data are presented in the “Results” section, in particular see Figure 10. The weight change was deterimed after single annealing run during 1 h at 1000°С and after step-wise heating from 200 to 1000°С. The results are discussed in the article. Following the recommendations of the Reviewer, we additionally performed XRD analysis of the oxidized coatings and added the results and discussion to the manuscript.